# Relevance of Mediterranean diet as a nutritional strategy in diminishing COVID-19 risk: A systematic review

Ceria Halim[1], Miranda Howen[1], Athirah Amirah Nabilah binti Fitrisubroto[1], Timotius Pratama[1], Indah Ramadhani Harahap[1], Lacman Jaya Ganesh[1], Andre Marolop Pangihutan Siahaan[2]*

1 Department of Center of Evidence Based Medicine, Faculty of Medicine, Universitas Sumatera Utara, Medan, North Sumatera, Indonesia, 2 Department of Neurosurgery, Faculty of Medicine, Universitas Sumatera Utara, Medan, North Sumatera, Indonesia

* andremarolop@usu.ac.id

**Data Availability Statement:** All relevant data are within the manuscript and its Supporting Information files.

## Abstract

### Background

Mediterranean Diet has been reported to possess immunomodulatory and anti-inflammatory properties. These properties are closely associated with the immunopathogenesis of COVID-19.

### Objective

The present systematic review aimed to determine the association between Mediterranean Diet and COVID-19, COVID-19 symptoms, and COVID-19 severity.

### Methods

The protocol for this systematic review was registered in International Prospective Register of Systematic Reviews (PROSPERO) with identification number CRD42023451794. The literature search was conducted through Pubmed, Proquest, and Google Scholar on August 2023. The inclusion criteria were studies with a population of human subjects, reported the association between Mediterranean diet adherence with risk of COVID-19 infection, COVID-19 symptoms, or COVID-19 severity, and full text must be available in English. The exclusion criteria were reviews, editorials, letters, replies, systematic reviews, meta-analyses, studies on animals, and duplicates. Risk of bias in included studies was assessed using Newcastle Ottawa Scale (NOS). Data was synthesized narratively. Each study was compared and a structured summary was developed.

### Results

After selection process, 6 articles were included, with a sample size of 55,489 patients. All studies were observational studies and assessed Mediterranean diet adherence using food frequency questionnaires (FFQ), with scoring system varied between each study. Four studies found a significant correlation between increased adherence to Mediterranean Diet and

**Funding:** The author(s) received no specific funding for this work.

**Competing interests:** The authors have declared that no competing interests exist.

reduced COVID-19 risk, while one study indicated non-significant association. One study reported a significant association between higher adherence to Mediterranean Diet and COVID-19 symptoms, but three studies reported non-significant association. One study found that individuals with higher adherence to Mediterranean Diet had reduced likelihood of developing severe COVID-19, however, two studies yielded inconclusive findings.

## Limitations

All studies used self-administrated food frequency questionnaires (FFQs), which were prone to biased responses, such as recall and estimation bias.

## Discussion

Lower trends of odds ratios (ORs) were consistently observed in higher Mediterranean diet adherence. In every outcome of the included studies, ORs ranged between 0.06–0.992, however, differing levels of significance were reported in each outcome.

## Conclusion

Overall analyses suggest that high adherence to Mediterranean Diet is a protective factor against COVID-19, with unclear benefits against COVID-19 symptoms and severity.

## Introduction

The coronavirus disease 2019 (COVID-19) was declared as a pandemic in March 2020 and had a widespread impact on individuals across the globe, with total infection cases reaching more than 775 million in 4 years [1, 2]. Studies have reported main symptoms of COVID-19 including fever, cough, and shortness of breath. Other non-specific symptoms were also reported among COVID-19 patients, such as sore throat, nasal congestion, and shortness of breath. In addition to the typical respiratory symptoms, COVID-19 has also been associated with a range of atypical manifestation, such as headache, diarrhea, nausea, and vomiting [3]. The World Health Organization (WHO) has classified COVID-19 disease severity into four groups: mild, moderate, severe, or critical. This classification was based on a combination of symptoms, oxygen saturation, and inflammatory biomarkers [3]. COVID-19 severity largely played a role in COVID-19 mortality, as reflected from recent data that reported COVID-19 mortality ranged from 6.3% in average population [4], to 13.2% in inpatient population, and drastically increased to 55.9% in patients requiring mechanical ventilation [5]. Until this article was written, more than 7 million total deaths by COVID-19 were recorded [2].

Inflammation has been widely recognized as a critical factor in the development and the degree of severity in COVID-19. After the initial innate immune response, a potent uncontrolled inflammatory response called cytokine storm may follow. Cytokine storm, characterized by excessive inflammation caused by high levels of cytokines, particularly IL-6, IL-1, and TNF-$\alpha$, can lead to advanced disease, multiorgan involvement, and serious consequences [6]. Cytokine storm has been identified as a potential target for therapeutic interventions, and multiple approaches. Immunomodulatory drugs, including dexamethasone and tocilizumab, as well as exosomes generated from mesenchymal stem cells and plasmapheresis, have been employed as ways to regulate this hyperinflammatory response [7]. These studies were just a few out of others that cemented the important role of inflammation in COVID-19.

The Mediterranean diet is a dietary pattern that consists primarily of plant-derived nutritional components, such as fruits, vegetables, legumes, nuts, and olive oil, and has been linked to anti-inflammatory properties and a reduced likelihood of developing cardiovascular disease [8]. Mediterranean diet is rich sources of polyphenols and polyunsaturated fatty acids (PUFAs), which are well-known antioxidants [9]. Adherence to Mediterranean diet has been reported to lower the expression of pro-inflammatory molecules, such as TNF-α, IL-1, IL-6 and CRP, as well as a reduction in the overall systemic inflammatory status, while also exerting antiviral effects by decreasing nuclear transcription factor kappa B (NF-κB) expression [10–12]. Mediterranean diet has been reported to benefit those with various disorders associated with persistent low-grade inflammation [10]. Researchers had suggested the potential association between diet and viral infections [13]. Fruits and fish oil, both vital components of Mediterranean diet, contain vitamin A, C, and D. These vitamins are effective antioxidant against reactive oxygen species (ROS) which are secreted by immune cells and help maintain body cells integrity, while also supporting epithelial barriers integrity. Vitamin D, which stimulate polymorphonuclear (PMN) and natural killer cells (NK) cells in producing potent anti-microbial peptides. The resulting effect are decreased risk of infection and increased viral clearance [13]. This has led to the growing proposition of the protective effects of Mediterranean diet towards COVID-19 [9, 11].

Based on the properties of Mediterranean diet that have been described above, this systematic review aims to address the association between Mediterranean diet and COVID-19 risk, COVID-19 symptoms, and COVID-19 severity.

## Materials and methods

### Protocol writing and registration

The protocol for this systematic review was registered on 15th September 2023 with the International Prospective Register of Systematic Reviews (PROSPERO) with identification number CRD42023451794. This systematic review was reported based on the Preferred Reporting Items for Systematic Reviews and Meta-Analyses (PRISMA) statement [14].

### Search strategy

Literature search was conducted on 3 search engines, Pubmed, Proquest and Google Scholar, on 16th August 2023. Literature search was done using the keyword "Mediterranean diet and COVID-19", with detailed search strategy and medical subject headings (MeSH) described below (S1 File). No additional article or abstract was selected from other sources.

### Study selection

The inclusion criteria for this systematic review were studies with a population of human subjects, and reported the association between Mediterranean diet adherence with COVID-19 infection, COVID-19 symptoms, or severe COVID-19. No study design nor publication date restriction were applied. Full-text must be available in English. The exclusion criteria were reviews, editorials, letters, replies, systematic reviews, meta-analyses, studies on animals, and duplicates.

Deduplication was carried out using the Rayyan application [15]. The remaining articles were screened based on title and abstract manually by two reviewers working individually. No automation tool was used in the process. Each article would be sought for further review if at least one reviewer considered it to fulfill the criteria and was within the scope of this systematic

review. Studies that passed the initial screening would be sought for retrieval. Two reviewers working together further reviewed the full-text of each literature for its eligibility.

### Data extraction

Data were collected by two reviewers working separately using data collection standards set previously. This included name of the first author, year of publication, study location, study design, total and characteristics of participants. Mediterranean diet association with COVID-19 infection, symptoms, and severity, Mediterranean diet adherence and outcome definition were also recorded. All results that were compatible with each outcome domain in each study were sought. Missing data were not sought further.

### Quality assessment

Non-randomized studies were assessed using Newcastle-Ottawa Scale (NOS) [16]. NOS criteria in cross-sectional studies were adapted from cohort criteria [17]. The NOS criteria were scored based on three categories: selection, comparability, and outcome, with each category consisted of 1–4 items. NOS score ranged from 0 (lowest quality) to 9 (highest quality). Randomized studies were assessed using revised Cochrane risk of bias tool for randomized trials (RoB2) [18]. Each successfully retrieved full-text article will be evaluated and scored by two reviewers working independently. Disagreement between reviewers' judgement was resolved by soliciting a third reviewer's opinion.

### Data synthesis

Data was synthesized narratively. The minimum number of studies was two for each analysis. The outcome was measured with odds ratio (OR) for COVID-19 infection, symptoms, and severity. Study characteristics, risk of bias, and study findings, and other relevant data were reported and tabulated. Similarities, differences, strengths, and limitations were compared across studies. A structured summary was also presented, to further elaborate the extracted data [19]. No subgroup nor sensitivity analyses were carried out.

### Reporting bias and certainty assessment

Reporting bias was evaluated using a tool by Page et al. [20] and RoBANS 2: A Revised Risk of Bias Assessment Tool for Nonrandomized Studies of Interventions [21]. We used the Grading of Recommendations, Assessment, Development, and Evaluations (GRADE) approach to define the certainty in the body of evidence [22].

## Results

### Literature search

A total of 325 records were identified across Pubmed, Proquest and Google Scholar. Duplicates were removed, with 209 records remained and screened. Thirteen articles were sought for retrieval and were further reviewed for eligibility. Six articles were deemed eligible and were included in this systematic review. A flow chart of selection process and exclusion reasons is provided below (Fig 1).

The study by Mohajeri et al. [23] appear to meet the inclusion criteria, but were excluded, as the outcome of COVID-19, COVID-19 symptoms, or COVID-19 severity were not reported. The study only reported outcome of inflammatory biomarkers.

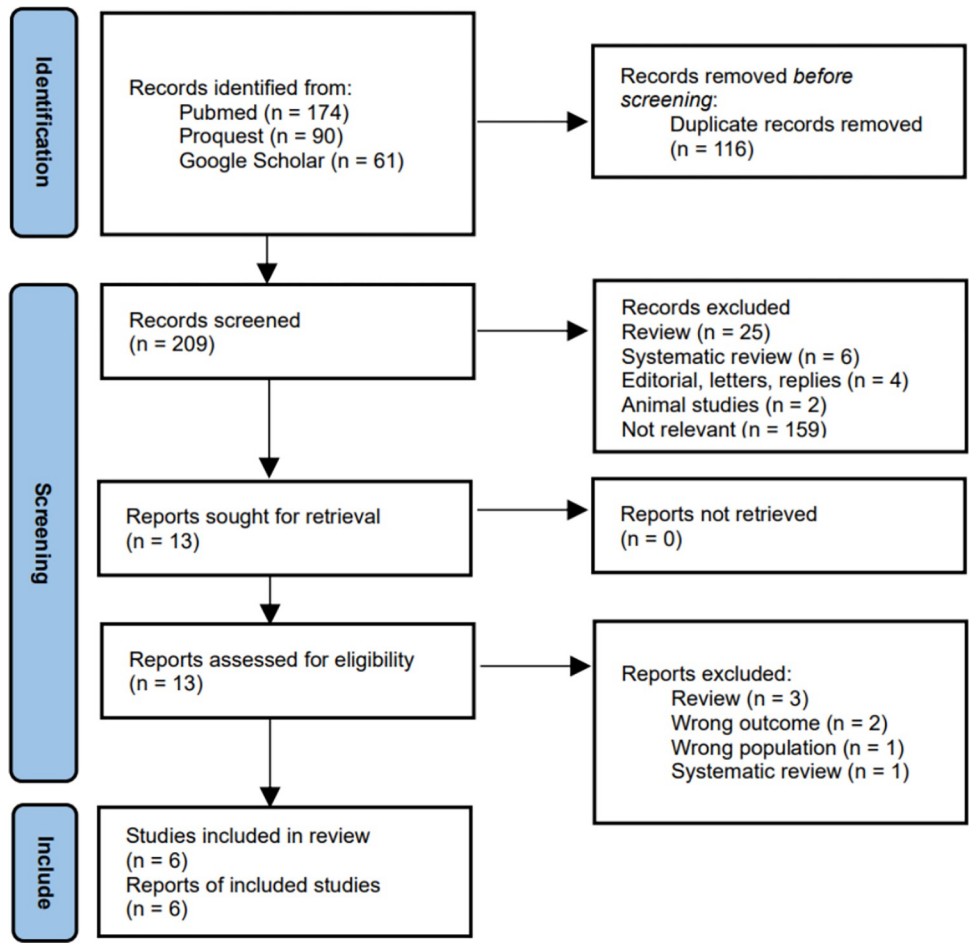

**Fig 1. Article selection process flow diagram.**

## Study characteristics

All 6 articles were observational studies, published in 2020–2023 [24–29]. The total sample was 55,489 patients, with study location across 5 countries. The majority of the studies had a prospective cohort design. Five studies reported the association between Mediterranean diet adherence and COVID-19 infection, four reported the association between Mediterranean diet adherence and COVID-19 symptoms, and three reported the association between Mediterranean diet adherence and severe COVID-19.

All studies measured the Mediterranean diet adherence using questionnaires, with scoring system varying between study to study. All Mediterranean diet adherence included categories such as, vegetables, fruits and nuts, cereals, legumes, fish, red meat, dairy, and alcohol consumption. Monounsaturated to saturated fat ratio was also an essential component in Mediterranean diet score, although four studies replaced monounsaturated fat to saturated fat ratio with olive oil intake [24–27]. The reason behind this was that the ratio of monounsaturated to saturated fats does not correspond to a specific food and the fact that olive oil is also the main source of monounsaturated fats in the traditional Mediterranean diet [25]. Newcastle Ottawa Scale (NOS), used to assess risk of bias was presented along the study characteristics in Table 1.

**Table 1. Study characteristics.**

| Author | Study Design | Study Location | Participants | Exposure | Food Groups | Outcome Analysed | NOS |
|--------|--------------|----------------|--------------|----------|-------------|------------------|-----|
| **El Khoury, 2021 [24]** | Observational case control | Lebanon | 399 Lebanese citizens residing in Lebanon, males or non-pregnant females, aged 21–64 years old | The MedDiet score, measured using FFQ, ranging from 0 (minimal adherence) to 55 (maximal adherence). | Score was calculated from 11 food groups: non-refined cereals, potatoes, fruits, vegetables, legumes, fish, red meat and products, poultry, full-fat dairy, olive oil, and alcohol | COVID-19 COVID-19 symptoms | 8 |
| **Perez-Araluce, R., 2022 [25]** | Prospective cohort | Spain | 9,485 participants | The Mediterranean Diet Score (MDS), measured using semiquantitative FFQ, ranging from 0 (minimal adherence) to 9 (maximal adherence). | Score was calculated from 9 categories: ethanol, olive oil, cereals, fruits and nuts, vegetables, legumes, fish, meat, whole dairy products. | COVID-19 COVID-19 symptoms COVID-19 severity | 8 |
| **Ponzo, V., 2021 [26]** | Observational retrospective | Italy | 900 health professionals still active at work and in contact with patients, aged 20–65 years old | The validated Medi-lite adherence score, measured using FFQ, ranging from 0 (minimal adherence) to 18 (maximal adherence). | Score was assessed from 9 food categories: fruit, vegetables, cereals, meat and meat products, dairy products, alcohol, and olive oil, legumes, and fish. | COVID-19 | 4 |
| **Sharma, S., 2023 [27]** | Prospective cohort | Italy | 1,520 participants | The Mediterranean Diet Score (MDS), measured using semi-quantitative FFQ. MDS ranging from 0 (minimal adherence) to 9 (maximal adherence). | Score was assessed from the 9 components: fruits and nuts, vegetables, legumes, fish, cereals, olive oil, meat, dairy products, ethanol consumption. | COVID-19 | 7 |
| **Yue, Y., 2022 [28]** | Prospective cohort | United States | 42,935 healthcare professionals aged 55–99 years old | The alternate Mediterranean Diet Score (aMED), measured using validated, self-administered, semiquantitative FFQ. AMED score ranging from 0 (minimal adherence) to 9 (maximal adherence). | Score was calculated from 9 components: vegetables, fruits, nuts, whole grains, legumes, fish, and the ratio of monounsaturated to saturated fat, red and processed meat consumption, and alcohol. | COVID-19 COVID-19 symptoms COVID-19 severity | 7 |
| **Zargarzadeh, N., 2022 [29]** | Retrospective cross-sectional | Iran | 250 patients aged 18–65 who had recovered from COVID-19 | The Mediterranean Diet Score (MDS), measured using FFQ. MDS ranging from 0 (minimal adherence) to 9 (maximal adherence). | Score was calculated from 9 categories: vegetables, legumes, fruits and nuts, cereal, and fish, meat, poultry and dairy, ethanol, and monounsaturated to saturated fat ratio | COVID-19 symptoms COVID-19 severity | 8 |

NOS = Newcastle Ottawa Scale; FFQ = Food Frequency Questionnaire; COVID-19 = Coronavirus Disease 2019.

## Mediterranean diet and COVID-19

A total of five studies investigated the association between Mediterranean diet adherence and COVID-19 [24–28] with a sample size of 55,239 patients as listed in Table 2. Three studies analyzed Mediterranean diet adherence as continuous variable of 1-point increment, one study analyzed Mediterranean diet adherence as categorical variable in a cut-off determined by the respective study, and one study analyzed Mediterranean diet adherence as both categorical variable in quartiles and continuous variable of 1-SD increment. Four studies relied on self-report of previous history of COVID-19 to determine COVID-19 cases, while one study relied on both serologic testing and self-report.

Three out of five studies reported Mediterranean diet adherence significantly lower risk of COVID-19, while two studies stated non-significant results. All five studies reported OR ranging between 0.75–0.948. The study by El Khoury et al., [24] reported the OR of being non infected by COVID-19 (OR = 1.055, 95%CI: 1.013–1.099, p = 0.01). This data was then inverted to properly reflect the odds of COVID-19 as noted in Table 2. Three studies adjusted for potential confounding variables. Three studies analyzed the association of Mediterranean

**Table 2. Mediterranean diet and COVID-19.**

| Author | Mediterranean Diet adherence | Comparison | COVID-19 definition | Results | Adjusted for | Mediterranean diet components analyzed |
|---|---|---|---|---|---|---|
| El Khoury, C. N., 2021 [24] | Mediterranean diet score, ranging from 0 to 55. | Each 1-point increment. | COVID-19 infection history is self reported. | Multiple binary logistic regression tests showed a significantly lower odds of COVID-19 (OR = 0.948, 95% CI: 0.910–0.987, p = 0.01). | Not stated. | None. |
| Perez-Araluce, R., 2022 [25] | Mediterranean Diet Score (MDS), ranging from 0 to 9. | High adherence group ($7 \leq MDS \leq 9$) and intermediate adherence group ($4 \leq MDS \leq 6$) were compared with low adherence group (MDS <4). | COVID-19 is self reported. Two definitions are used. One, more specific, participants who reported a positive diagnostic test. Two, a broader definition, participants with a medical diagnosis were also included. | High adherence group had insignificant lower odds for positive diagnostic COVID-19 test (OR = 0.75, 95% CI: 0.49–1.13, p for trend = 0.166) and both positive diagnostic test and medical diagnosis of COVID-19 (OR = 0.79, 95% CI: 0.58–1.08, p for trend = 0.137), when compared to low adherence group. | Age, sex, years of university studies, marital status, smoking, BMI, physical activity at leisure-time, year of entering the cohort, previous diagnosis of chronic diseases, and category of consumption of the other food groups. | Yes. Higher olive oil and lower red meat consumption showed lower odds of COVID-19 risk. |
| Ponzo, V., et al., 2021 [26] | Medi-lite adherence score, ranging from 0 to 18. | Each 1-point increment. | SARS-COV-2 infection is self reported. | In a logistic regression model, each 1-point increment in the MeD score was associated with a lower risk of SARS-COV-2 infection (OR = 0.88, 95% CI: 0.81–0.97, p = 0.01). | Not stated. | Yes. Higher cereal consumption lower SARS-COV-2 infection. |
| Sharma, S., 2023 [27] | Mediterranean diet score, ranging from 0 to 9. | Each 1-point increment. | SARS-CoV-2 infection cases were determined through serology, and previous clinical diagnosis of COVID-19 disease was self-reported. | In multivariable-adjusted logistic regression models, increased adherence to a traditional MD was not associated with risk of SARS-CoV-2 infection (OR = 0.94, 95% CI: 0.83–1.06; p-value for trend = 0.30) | Age, sex, energy intake, educational level, occupational class, marital status, being health professional, smoking status, body mass index, leisure-time physical activity, baseline history of CVD, cancer, diabetes, hypertension, dyslipidemia, number of chronic diseases diagnosed since March 2020, and the composite index of behavioral and environmental risk factors SARS-CoV-2 infection. | Yes. Higher consumption of olive oil, moderate amounts of alcohol, and higher intake of fruits and nuts were associated with lower odds of COVID-19. |
| Yue, Y., et al., 2022 [28] | The alternative Mediterranean diet (AMED) score, ranging from 0 to 9. | The multivariable-adjusted ORs comparing the top with the bottom quartile of AMED score. Continuous analyses for a 1-SD increment were also reported. | Self-reported SARS-CoV-2 infection, including positive results from an antigen or antibody test. | Using multivariable logistic regression models, the top quartile AMED had lower risk of SARS CoV-2 infection than the bottom quartile (OR = 0.78, 95% CI: 0.67–0.92; P-trend = 0.0032) Continuous analyses for 1-SD increment showed similar results (OR = 0.90, 95% CI: 0.85–0.95, p-trend = 0.0032) | Age, sex, race, smoking, physical activity, total energy intake, census tract median family income, census tract median family home value, census tract population density, concern about COVID-19, interaction with people other than patients with presumed or documented COVID-19, frontline health care providers and PPE use, BMI, history of high cholesterol, history of high blood pressure, and presence of other pre-existing medical conditions. | None. |

OR = odds ratio; CI = confidence interval; SD = standard deviation; COVID-19 = Coronavirus Disease 2019; SARS-CoV-2 = Severe Acute Respiratory Syndrome Coronavirus 2; BMI = body mass index; PPE = personal protective equipment.

diet individual components with COVID-19 risk. Among the food categories, higher olive oil consumption, lower red meat consumption, lower cereal consumption, moderate amounts of alcohol, and higher intake of fruit and nuts were reported to lower risk against COVID-19.

## Mediterranean diet and COVID-19 symptoms

Four studies reported the association between Mediterranean diet adherence and COVID-19 symptoms [24, 25, 28, 29] with a sum of 53,069 patients. Differing comparison of Mediterranean diet adherence score was reported across studies with one each of, continuous variable of 1-point increment, categorical variable in a cut-off determined by the authors of the study, categorical variable in tertiles, and as both categorical variable in quartiles and continuous variable of 1-SD increment, as detailed in Table 3. All studies depended on self-report for the

**Table 3. Mediterranean diet and COVID-19 symptoms.**

| Author | Mediterranean Diet adherence | Comparison | COVID-19 symptoms definition | Results | Adjusted for | Mediterranean diet components analysed |
|---|---|---|---|---|---|---|
| El Khoury, C. N., 2021 [24] | Mediterranean diet score, ranging from 0 to 55. | Each 1-point increment. | Self-reported COVID-19 symptoms. The COVID-19 burden was determined based on the number of symptoms and hospitalization as follows; <5 symptoms: mild burden, between 5 and 10 symptoms: moderate burden. | Using multiple adjusted logistic regression, MedDiet score did not reduce the risk of moderate COVID-19 burden, (OR = 0.992; 95% CI: 0.873–1.129, p = 0.908). | Age, pre-existing health conditions and respiratory diseases | None. |
| Perez-Araluce, R., 2022 [25] | Mediterranean Diet Score, ranging from 0 to 9. | High adherence group (7 ≤ MDS ≤ 9) and intermediate adherence group (4 ≤ MDS ≤ 6) were compared with low adherence group (MDS <4). | Self-reported cases of symptomatic COVID-19, with symptoms included in this definition were: cough, cold, respiratory distress, loss of smell or taste, diarrhea, and fever | Using multivariable adjusted logistic regression models, high adherence group had insignificant lower odds for symptomatic COVID-19 (OR = 0.84; 95% CI: 0.60–1.16, P = 0.294) | Age, sex, years of university studies, marital status, smoking, BMI, physical activity at leisure-time, year of entering the cohort, previous diagnosis of chronic diseases, and category of consumption of the other food groups. | Yes. No components were found to be significant. |
| Yue, Y., et al., 2022 [28] | The alternative Mediterranean diet (AMED) score, ranging from 0 to 9. | The multivariable-adjusted ORs comparing the top with the bottom quartile of AMED score. Continuous analyses for a 1-SD increment were also reported. | Self-reported COVID-19 symptoms including fever, sore throat, muscle aches, loss of taste, loss of smell, and other symptoms consistent with COVID-19 infection. | Using multivariable logistic regression models, the association between AMED score and symptomatic SARS-CoV 2 infection reached borderline significance with lower risk of symptomatic SARS-CoV 2 infection trends (OR = 0.89, 95% CI: 0.80–0.99; P-trend = 0.0549) Continuous analyses for 1-SD increment showed similar results (OR = 0.96, 95% CI: 0.93–1.00, p-trend = 0.0549) | Age, sex, race, smoking, physical activity, total energy intake, census tract median family income, census tract median family home value, census tract population density, concern about COVID-19, interaction with people other than patients with presumed or documented COVID-19, frontline health care providers and PPE use, BMI, history of high cholesterol, history of high blood pressure, and presence of other pre-existing medical conditions. | None. |

*(Continued)*

**Table 3.** (Continued)

| Author | Mediterranean Diet adherence | Comparison | COVID-19 symptoms definition | Results | Adjusted for | Mediterranean diet components analysed |
|---|---|---|---|---|---|---|
| Zargarzadeh, N., 2022 [29] | Mediterranean diet score, ranging from 0 to 9. | Participants were divided into tertile groups based on their MD score. | Self-reported presence of common COVID-19 manifestations, i.e., fever, chilling, dyspnea, cough, weakness, muscle pain, sore throat, nausea, and vomiting. | Using binary logistic regression, there was a significant inverse relationship between the Mediterranean diet score and the likelihood of experiencing COVID-19 symptoms such as dyspnea (OR = 0.32; 95% CI: 0.13–0.76, P = 0.03), cough (OR = 0.11; 95% CI: 0.05–0.26, P <0.001), fever (OR = 0.11; 95% CI: 0.03–0.37, P <0.001), chilling (OR = 0.06; 95% CI: 0.01–0.26, P <0.001), weakness (OR = 0.34; 95% CI: 0.15–0.76, P = 0.01), myalgia (OR = 0.34; 95% CI: 0.16–0.72, P = 0.005), nausea and vomiting (OR = 0.06; 95% CI: 0.01–0.30, P <0.001), and sore throat (OR = 0.08; 95% CI: 0.03–0.21, P <0.001). | Age, sex, energy intake/BMR, physical activity, supplement use, corticosteroids use, antiviral drugs use, BMI. | None. |

OR = odds ratio; CI = confidence interval; SD = standard deviation; COVID-19 = Coronavirus Disease 2019; BMI = body mass index; PPE = personal protective equipment; BMR = basal metabolic rate.

presence of COVID-19 symptoms, and analyzed the odds ratio using logistic regression with yes/no type outcome. Two studies analyzed for symptomatic COVID-19 outcome [25, 28], one study for moderate COVID-19 burden (defined as presence of 5–10 symptoms) [24], and one study analyzed for each COVID-19 symptom [29].

The study by Zargarzadeh et al. [29] found that higher Mediterranean diet adherence significantly decrease the odds for all reported COVID-19 symptoms, with OR varied between 0.06–0.34 for each symptom. Three studies reported insignificant association, with OR of each study ranging between 0.84–0.992, and p value >0.05, although the result from Yue et al. [28] did reach the limit of statistical significance (OR = 0.89, 95% CI: 0.80–0.99; P-trend = 0.0549). Perez-Araluce et al. analyzed the relation of individual components of Mediterranean diet with COVID-19 symptoms, yet no food categories were found to be significant [25].

## Mediterranean diet and COVID-19 severity

The association between Mediterranean diet adherence and COVID-19 severity were analyzed in three observational studies [25, 28, 29], as listed in Table 4. The total sample was 52,670 patients, relatively similar with the previous two outcomes. Self-report of hospitalization due to COVID-19 was the main method to determine severe COVID-19 cases. One study examined the medical records of the participants for severe COVID-19, in accordance with the National Institute of Health's Coronavirus Disease 2019 (COVID 19) Treatment Guidelines.

The study by Zargarzadeh et al. [29] found that participants with top tertile Mediterranean diet score were less likely to have severe COVID-19 (OR = 0.23; 95% CI: 0.11–0.50, P <0.001) than the bottom tertile. Two studies reported insignificant association with OR ranging 0.22–0.89, and p value > 0.05. Two studies analyzed the effects of Mediterranean diet individual components against COVID-19 severity. One study found no individual components were significant, while the study by Zargarzadeh et al. [29] reported higher consumption of vegetables, fruits, legumes, nuts, whole grains, and fish lower odds of severe COVID-19.

**Table 4. Mediterranean diet and COVID-19 severity.**

| Author | Mediterranean Diet adherence | Comparison | Severe COVID-19 definition | Results | Adjusted for | Mediterranean diet components analysed |
|---|---|---|---|---|---|---|
| **Perez-Araluce, R., 2022 [25]** | Mediterranean Diet Score (MDS), ranging from 0 to 9. | High adherence group (7 ≤ MDS ≤ 9) and intermediate adherence group (4 ≤ MDS ≤ 6) were compared with low adherence group (MDS <4). | Self-reported cases of COVID-19 that required hospitalization with symptoms compatible with the disease. | Multivariable adjusted, higher adherence was not associated with severe COVID-19 (OR = 0.22; 95% CI: 0.03–1.77, p for trend = 0.153) compared to low adherence group. | Age, sex, years of university studies, marital status, smoking, BMI, physical activity at leisure-time, year of entering the cohort, previous diagnosis of chronic diseases, and category of consumption of the other food groups. | Yes. No components were found to be significant. |
| **Yue, Y., et al., 2022 [28]** | The alternative Mediterranean diet (AMED) score, ranging from 0 to 9. | Continuous analyses ORs for a 1-SD increment in AMED score. | Self-reported COVID-19 participants that required hospitalization. | Using multivariable logistic regression models, 1-SD increments of AMED scores was not associated with severe infection (OR = 0.89, 95% CI: 0.73–1.08; P-trend = 0.8288) | Age, sex, race, smoking, physical activity, total energy intake, census tract median family income, census tract median family home value, census tract population density, concern about COVID-19, interaction with people other than patients with presumed or documented COVID-19, frontline health care providers and PPE use, BMI, history of high cholesterol, history of high blood pressure, and presence of other pre-existing medical conditions. | None. |
| **Zargarzadeh, N., 2022 [29]** | Mediterranean diet score (MDS), ranging from 0 to 9. | Participants were divided into tertile groups based on their MDS. | The researchers examined the medical records of the participants, with severe COVID-19 classified using the National Institute of Health's Coronavirus Disease 2019 (COVID 19) Treatment Guidelines. | In multivariable-adjusted logistic regression models, highest MDS was 77% less likely to have severe COVID-19 than those with the lowest score (OR = 0.23; 95% CI: 0.11–0.50, P <0.001). | Age, sex, energy intake/BMR, physical activity, supplement use, corticosteroids use, antiviral drugs use, BMI. | Yes. Higher consumption of vegetables, fruits, legumes, nuts, whole grains, and fish were reported to lower odds of severe COVID-19. |

OR = odds ratio; CI = confidence interval; SD = standard deviation; COVID-19 = Coronavirus Disease 2019; BMI = body mass index; PPE = personal protective equipment; BMR = basal metabolic rate.

## Reporting bias and evidence grading

Using the tool by Page et al. [20] and RoBANS 2 [21] for evaluating reporting bias, we considered the included studies were of low-to-moderate risk for reporting bias. A detailed assessment is attached (S2 File). To grade the quality of evidence, the tool Grading of Recommendations, Assessment, Development, and Evaluations (GRADE) was used. On the outcome of COVID-19 risk, the authors considered the overall grading of the evidence to be of moderate certainty, meaning the true effect is probably close to the estimated effect. This was primarily because of the large number of patients for the outcome analyzed, similar effect estimates trend between studies (all outcome analyses reported OR <1), and adjustment for multiple confounding factors. For the outcome of COVID-19 symptoms and COVID-19 severity,

we rated the evidence to be of low certainty, meaning the true effect might be markedly different from the estimated effect. Certainty for these outcomes was rated down for imprecision and that there were too few available studies included in the analyses.

## Discussion

The protective mechanism of diet against viral respiratory disease progression has been acknowledged by several studies. A previous study found that healthy plant-based foods was associated with lower odds and severity of COVID-19 [30]. Another relevant finding was that the adoption of a Traditional Mediterranean Diet contributed towards to the improvement of patients with recurring colds and frequent inflammatory complications, with significantly reduced episodes and symptoms [31]. A previous systematic review found that Mediterranean diet was reported to lower inflammatory biomarker levels in obese/overweight adults [32]. Similarly, a precedent meta-analysis reported Mediterranean diet adherence effectively reduce SARS-CoV-2 infection by 78% (95% CI 69%–88%), although the authors recommended cautious interpretation due to the paucity of the included studies [33].

These previous studies were in line with our findings. For the outcome of COVID-19, interestingly, all studies reported OR<1, although there were differing reports on the significance of this association. Another interesting thing to note was the studies that didn't adjust for confounding factors reported significant association, while two studies that adjusted for confounding factors reported non-significant association. An exception to this was the study with the largest number of participants. The study by Yue et al. [24] adjusted for multiple confounding factors and reported higher Mediterranean diet adherence proved to be significant in lowering odds of COVID-19. These results indicated that higher Mediterranean diet adherence was a protective factor against COVID-19, although the magnitude of the effect estimate was probably minor and thus was only observable in a larger population.

For the outcomes of COVID-19 symptoms and severity, all studies were adjusted for confounding variables, and almost all studies recorded non-significant association. Only the study by Zargarzadeh et al. [29] reported significant association for both outcomes. These differences could be explained by a few reasons. First, the study had a cross-sectional design, and as stated by its authors, the results cannot infer a causative association. In addition, although the study scored a high NOS score, this score was obtained by using a form adapted for cross-sectional study, and may not be comparable to the NOS from cohort studies. It bears mentioning that, despite the difference in significance, the effect estimates for both COVID-19 symptoms and severity outcomes across all studies showed a consistent trend of OR <1. It was necessary to put into consideration that the consistent effect of Mediterranean diet adherence in all studies may be more critical than the statistically insignificant p-value [34]. In fact, a nonsignificant result does not mean that there is no effect [34], albeit it might be too presumptive to assume a protective effect of higher Mediterranean diet adherence against COVID-19 symptoms and severity without conclusive evidence. Although a meta-analysis could produce a higher statistical power for this purpose, there were a few considerations as to why it was not performed. Multiple assessment index for Mediterranean diet adherence were used between studies. In addition, most studies assigned points for Mediterranean diet adherence depending on sex-specific median consumption in the respective study. Hence, we deemed a meta-analysis was not appropriate for the diverse data.

Several subtypes of food were inversely associated with COVID-19 risk. Higher olive oil consumption, lower red meat consumption, lower cereal consumption, moderate amounts of alcohol, and higher intake of fruit and nuts reduced COVID-19 risk, and higher consumption of vegetables, fruits, legumes, nuts, whole grains, and fish lowered odds of severe COVID-19.

The presence of flavonoids, an antiviral and immunostimulatory compounds, linked fruit consumption to a reduced risk of SARS-CoV-2 infection [35]. Vitamin C, a major vitamin in fruits and vegetables, took part in inhibiting NLRP3 inflammasome pathway, decreasing pro-inflammatory cytokines, and improving neutrophil chemotaxis [35, 36]. These resulted in minimizing viral pathogenesis, enhancing recovery, while also preventing respiratory viral infection [36, 37]. Olive oil and fish were excellent sources of monounsaturated fatty acids (MUFAs) and polyunsaturated fatty acids (PUFAs), which had been known to exhibit immunomodulatory properties [38]. Docosahexaenoic acid (DHA) and eicosapentaenoic acid (EPA) bind to peroxisome proliferator-activated receptors (PPARs), inhibiting cytokine expression by NF-κB inflammatory transcription factor [38]. Other pathways include decreasing lipid rafts and augmenting major histocompatibility complex 1 (MHC 1) expression [38]. In addition, virgin olive oil contained phenolic compounds, such as oleocanthal, that exerted potent anti-inflammatory actions, similar to ibuprofen [39]. Likewise, legumes possess bioactive compounds, such as peptides, polyphenols and saponins, which showed anti-inflammatory activity, inhibiting cyclooxygenase-1 (COX-1) and cyclooxygenase-1 (COX-2) enzyme expression [40]. Red wine, a prevalent alcoholic drink in Mediterranean diet, contains many types of antioxidants which neutralize free oxygen radicals released by neutrophils, minimizing cellular damage [41]. The overall beneficial anti-inflammatory properties of these food groups explained their effects in diminishing risk of COVID-19 and severe COVID-19.

## Strengths and limitations

We further reviewed the strengths and limitations of each study, as listed in Table 5. The study by Sharma et al. used objective serology data coupled with self-reported data to define COVID-19 cases [27]. The study by Yue et al. had a large number of participants, and all participants were health professionals which ensured high-quality health information [28]. Most of the studies included were adjusted for multiple confounding variables.

The major limitation was that Mediterranean diet adherence of all studies was self-reported. Measurement errors in FFQs are known to affect results of studies and may have biased the observed effect estimates. FFQs were prone to biased responses, such as recall and estimation bias. However, FFQs were often used in large cohort studies because it is realistic and is logistically feasible. Two of the studies stated they had inadequate cases for observed outcome to produce a significant result [27, 28].

To our knowledge, this systematic review is the first to specifically investigate the relationship between Mediterranean diet and COVID-19 risk, symptoms, and severity. Additionally, each of the study selection and bias assessment process was conducted by multiple reviewers to ensure minimal bias. This was done in accordance with the Cochrane guidelines [42].

This systematic review has some limitations that should be considered when interpreting the results. First, there were few studies included in this systematic review. Second, as a meta-analysis is not performed, the magnitude of the effects could not be precisely measured.

## Public health implications

This systematic review provides an up-to-date summary of the available evidence. As more and more countries have loosened on personal protective equipment (PPE) and social distancing regulations, a nutritional strategy may be more feasible and beneficial long-term. The results of the present study may shed some light on additional benefits of Mediterranean diet against COVID-19. The findings also suggest that specific food groups in the Mediterranean diet may be more important in reducing COVID-19 odds. More studies should be conducted before definitive conclusions can be drawn.

**Table 5. Strengths and limitations of each study.**

| Studies | Strengths | Limitations |
|---|---|---|
| **El Khoury, C. N., 2021 [24]** | The simultaneous use of an a priori and two a posteriori methods allowed a complementary evaluation of the dietary intake. | • Observational study.<br>• Limited participation of COVID-19 cases.<br>• Self-administrated questionnaire with a possibility of a recall and estimation bias.<br>• Not assessing for other aspects that affect the occurrence and the outcomes of the infection. |
| **Perez-Araluce, R., 2022 [25]** | - | • Focused on non-healthcare participants.<br>• Unable to measure non-pharmacological preventive measures. |
| **Ponzo, V., et al., 2021 [26]** | The first that specifically evaluated the association between Mediterranean diet adherence and SARS-COV-2 infection. | • The design of the study did not allow for establishing a direct relationship.• Many confounding factors related to sociocultural factors, government response to the pandemic, and differences in healthcare systems across countries may have influenced the results.• Questionnaires did not properly estimate the micronutrient intakes of participants.• The enrolment of healthcare professionals from a single region and the low percentage of male participants, as there is a gender bias in participating in the completion of food questionnaires. |
| **Sharma, S., 2023 [27]** | • Well-established cohort, participants were followed up during the pandemic.<br>• Using objective serology data coupled with self-reported data—which increases the robustness of the study methodology of ascertaining the SARS-CoV-2 cases.<br>• Detailed dietary information, including macronutrients and micronutrients.<br>• The regression models were carefully adjusted for confounders<br>• Included health professionals in the sub-cohort which increase the generalizability. | • Dietary intakes were estimated based on participants memory recall.<br>• FFQ could have led to bias and under/over-estimation of the dietary intakes.<br>• Relatively low number of infection cases. |
| **Yue, Y., et al., 2022 [28]** | • Had access to validated and repeated measures of long-term diet.<br>• Information on SARS-CoV-2 infection and symptoms were captured in a timely manner.<br>• All study participants were health professionals which allowed authors to capture high-quality health information. | • Only invited participants who completed the most recent survey to the COVID-19 study, which could lead to selection bias.<br>• No information on fatal COVID-19 cases and outcomes were based on self-report data.<br>• Only a small number of hospitalized COVID-19 cases.<br>• Possibility of residual confounding due to socioeconomic, lifestyle, and health conditions cannot be ruled out given the observational nature of the study. |
| **Zargarzadeh, N., 2022 [29]** | Shed light on a nutritional strategy that could be used in the fight against the COVID-19 pandemic. | • Cross-sectional design, this study cannot infer causation, and thus recommendations cannot be affirmed.<br>• Possibility of residual confounding cannot be completely ruled out.<br>• All participants were drawn from a single-center, so generalizability to the general public should be approached with caution.<br>• FFQs rely on interviewees' memory, the memory bias inherent in this type of questionnaire cannot be ignored. |

## Conclusion

Overall, the analyses suggest higher Mediterranean diet adherence significantly reduced odds of COVID-19, with non-significant results against COVID-19 symptoms and severity.

## Supporting information

**S1 Checklist. PRISMA 2020 abstract checklist.**
(DOCX)

**S2 Checklist. PRISMA 2020 main checklist.**
(DOCX)

**S1 File. Detailed search strategies and MeSH.**
(DOCX)

**S2 File. Reporting bias risk assessment.**
(DOCX)

## Acknowledgments

We sincerely thank all researchers who have contributed their studies, without whom this systematic review would not be possible.

## Author Contributions

**Conceptualization:** Ceria Halim.

**Data curation:** Ceria Halim, Miranda Howen, Athirah Amirah Nabilah binti Fitrisubroto, Timotius Pratama, Indah Ramadhani Harahap, Lacman Jaya Ganesh.

**Formal analysis:** Ceria Halim, Miranda Howen, Athirah Amirah Nabilah binti Fitrisubroto, Timotius Pratama, Indah Ramadhani Harahap, Lacman Jaya Ganesh.

**Funding acquisition:** Andre Marolop Pangihutan Siahaan.

**Investigation:** Ceria Halim, Miranda Howen, Athirah Amirah Nabilah binti Fitrisubroto.

**Methodology:** Ceria Halim.

**Project administration:** Andre Marolop Pangihutan Siahaan.

**Resources:** Ceria Halim.

**Supervision:** Andre Marolop Pangihutan Siahaan.

**Validation:** Timotius Pratama, Indah Ramadhani Harahap, Lacman Jaya Ganesh.

**Visualization:** Ceria Halim.

**Writing – original draft:** Ceria Halim, Timotius Pratama, Lacman Jaya Ganesh.

**Writing – review & editing:** Ceria Halim, Andre Marolop Pangihutan Siahaan.

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
