## [Decision Letter · Decision Letter 0]

24 Jun 2024

PONE-D-24-10878Relevance of Mediterranean diet as a nutritional strategy in diminishing COVID-19 risk: A systematic reviewPLOS ONE

Dear Dr. Siahaan,

Thank you for submitting your manuscript to PLOS ONE. After careful consideration, we feel that it needs minor revision to fully meet PLOS ONE’s publication criteria as it currently stands. Therefore, we invite you to submit a revised version of the manuscript that addresses the points raised during the review process.

Please submit your revised manuscript within Aug 08 2024 11:59PM. If you will need more time than this to complete your revisions, please reply to this message or contact the journal office at plosone@plos.org. Please include the following items when submitting your revised manuscript:A rebuttal letter that responds to each point raised by the academic editor and reviewer(s). You should upload this letter as a separate file labeled 'Response to Reviewers'.A marked-up copy of your manuscript that highlights changes made to the original version. You should upload this as a separate file labeled 'Revised Manuscript with Track Changes'.An unmarked version of your revised paper without tracked changes. You should upload this as a separate file labeled 'Manuscript'.If applicable, we recommend that you deposit your laboratory protocols in protocols.io to enhance the reproducibility of your results. Protocols.io assigns your protocol its own identifier (DOI) so that it can be cited independently in the future. For instructions see: https://journals.plos.org/plosone/s/submission-guidelines#loc-laboratory-protocols. Additionally, PLOS ONE offers an option for publishing peer-reviewed Lab Protocol articles, which describe protocols hosted on protocols.io. Read more information on sharing protocols at https://plos.org/protocols?utm_medium=editorial-email&utm_source=authorletters&utm_campaign=protocols.

We look forward to receiving your revised manuscript.

Kind regards,

Nour Amin Elsahoryi, pHD

Academic Editor

PLOS ONE

Journal Requirements:

Reviewers' comments:

Reviewer's Responses to Questions

**Comments to the Author**

1. Is the manuscript technically sound, and do the data support the conclusions?

Reviewer #1: Yes

Reviewer #2: Yes

2. Has the statistical analysis been performed appropriately and rigorously? 

Reviewer #1: Yes

Reviewer #2: I Don't Know

3. Have the authors made all data underlying the findings in their manuscript fully available?

Reviewer #1: Yes

Reviewer #2: Yes

4. Is the manuscript presented in an intelligible fashion and written in standard English?

Reviewer #1: Yes

Reviewer #2: Yes

5. Review Comments to the Author

Reviewer #1: General comment:

The topic approached by authors of this paper is interesting and overall, I believe the study has relevant information hereby suitable for publication after revision.

The authors clearly explained their work throughout the manuscript. I just have a suggestion to point out.

The paper is generally well-written and easy to follow. However, certain sections, particularly the results and discussion, could benefit from more concise language and avoidance of repetition.

Specific comments:

Introduction:

- It could benefit from a more detailed explanation of the specific mechanisms through which the Mediterranean diet might influence COVID-19 outcomes.

- To strengthen the context, it would be good to include more recent studies and comprehensive statistics on the global impact of COVID-19.

Methods:

- Specific search terms and possible language restrictions could be included in the search strategy.

- More details on the scoring process of the Newcastle Ottawa Scale (NOS) and how the authors addressed any biases identified would enhance the reader's understanding of the robustness of the findings.

- The cross-sectional design of the studies included in this review limits the ability to establish causation.

- There is a limited number of studies included in this review.

Results:

- The narrative synthesis of the data is clear, but the inclusion of a meta-analysis could provide a more quantifiable measure of the impact of the Mediterranean diet.

Discussion:

- Further discussion on the biological plausibility of the Mediterranean diet’s effects on immune function and inflammation in the context of viral infections would be beneficial.

Reviewer #2: The manuscript is well written. In the introduction, the author explained the reason for the scientific gap very well. Also, the Material and Method section and the discussion are detailed and precise. Overall, well-conducted study.

6. PLOS authors have the option to publish the peer review history of their article (what does this mean?). If published, this will include your full peer review and any attached files.

Reviewer #1: No

Reviewer #2: No

---

## [Author Response · Author response to Decision Letter 0]

20 Jul 2024

Dear reviewers, we appreciate your time and attention in reviewing the manuscript and giving us your valuable inputs. We have made several changes throughout the manuscript to better accommodate your suggestions. We have attached a Response to Reviewers file for further details. Please do not hesitate to contact us for further queries. Thank you kindly.

---

## [Editor Report · Decision Letter 1]

23 Jul 2024

Relevance of Mediterranean diet as a nutritional strategy in diminishing COVID-19 risk: A systematic review

PONE-D-24-10878R1

Dear Dr.  Andre 

We’re pleased to inform you that your manuscript has been judged scientifically suitable for publication and will be formally accepted for publication once it meets all outstanding technical requirements.

Kind regards,

Nour Amin Elsahoryi, pHD

Academic Editor

PLOS ONE

---

## [Editor Report · Acceptance letter]

29 Jul 2024

PONE-D-24-10878R1 

PLOS ONE

Dear Dr. Siahaan, 

I'm pleased to inform you that your manuscript has been deemed suitable for publication in PLOS ONE. Congratulations! Your manuscript is now being handed over to our production team.

Kind regards, 

on behalf of

Dr. Nour Amin Elsahoryi 

Academic Editor

PLOS ONE